# The Dual Role of Necroptosis in Pancreatic Ductal Adenocarcinoma

**DOI:** 10.3390/ijms241612633

**Published:** 2023-08-10

**Authors:** Valentina Giansante, Gianmarco Stati, Silvia Sancilio, Emanuela Guerra, Saverio Alberti, Roberta Di Pietro

**Affiliations:** 1Department of Medicine and Aging Sciences, Section of Biomorphology, “G. d’Annunzio” University of Chieti-Pescara, 66100 Chieti, Italy; 2Laboratory of Cancer Pathology, Center for Advanced Studies and Technologies (CAST), “G. d’Annunzio” University of Chieti-Pescara, 66100 Chieti, Italy; 3Department of Medical, Oral and Biotechnological Sciences, “G. d’Annunzio” University of Chieti-Pescara, 66100 Chieti, Italy; 4Unit of Medical Genetics, Department of Biomedical Sciences, University of Messina, 98122 Messina, Italy; 5Sbarro Institute for Cancer Research and Molecular Medicine, Center for Biotechnology, Department of Biology, College of Science and Technology, Temple University, Philadelphia, PA 19122, USA

**Keywords:** pancreatic cancer, necroptosis, tumor microenvironment, cell death

## Abstract

Pancreatic cancer (PC) is the seventh leading cause of cancer-related death. PC incidence has continued to increase by about 1% each year in both men and women. Although the 5-year relative survival rate of PC has increased from 3% to 12%, it is still the lowest among cancers. Hence, novel therapeutic strategies are urgently needed. Challenges in PC-targeted therapeutic strategies stem from the high PC heterogeneity and from the poorly understood interplay between cancer cells and the surrounding microenvironment. Signaling pathways that drive PC cell growth have been the subject of intense scrutiny and interest has been attracted by necroptosis, a distinct type of programmed cell death. In this review, we provide a historical background on necroptosis and a detailed analysis of the ongoing debate on the role of necroptosis in PC malignant progression.

## 1. Introduction

Pancreatic Cancer (PC) currently is the seventh leading cause of cancer-related death [1] and is expected to become the second one by 2030 [2].

PC incidence differs widely among countries: the highest age-standardized incidence is seen in Europe and North America, the lowest is recorded in Africa and South-Central Asia [3]. 

Late diagnosis, together with chemoresistance and aggressive biology, have contributed to its poor prognosis and to the concept of PC as a “silent killer”. 

Early clinical manifestations are often vague, depending on the site of origin of the tumor. Approximately 70% of pancreatic tumors arise in the head of the pancreas and cause biliary obstruction leading to dark urine (49%) and jaundice (49%). Patients with PC arising from the pancreas body or tail present other symptoms, such as abdominal pain, back pain, and cachexia-related symptoms (appetite loss, weight loss, fatigue) [4].

Pancreatic Ductal Adenocarcinoma (PDAC) is the most prevalent type of PC, representing more than 90% of cases [5]. It shows an overall 5-year survival rate of less than 5% and a median survival time of less than 6 months if untreated [6]. 

Still today, there is a common and keen interest in filling the gaps of our knowledge about PC due to recent interesting findings that have encouraged researchers to go deeply into the complexity of this lethal disease.

The characteristic aggressiveness of PC should probably be assigned to several undisclosed strategies developed by neoplastic cells, that could be enhanced by the same pancreatic tissue context, particularly hostile but also crucial to cause a more powerful response by the tumor itself.

In this perspective, we discuss the recent hypothesis that necroptosis, a caspase-independent mechanism of programmed cell death, may play an unexpected role in PDAC malignant progression through a peculiar interaction between the tumor cell and the extra-epithelial compartment. Here, we review opposing views on the contribution of programmed cell death in tumor progression and on the role of tumor microenvironment as a final “tiebreaker”.

## 2. Pancreatic Ductal Adenocarcinoma (PDAC)

After colorectal cancer, PDAC is the second most common cancer of the digestive system in the United States [7]. Only 10% of PDAC cases appear related to family history. Inherited cancer syndromes with known germline mutations, such as the Lynch syndrome (MLH1, MLH2, MLH6, PMS2), familiar breast and ovarian cancer (BRCA1 and BRCA2), familial adenomatous polyposis (FAP), familial atypical multiple mole melanoma (CDKN2A), and Peutz–Jeghers (STK11/LKB1), show an increased risk of PDAC [8]. 

In total, 95% of PDAC show activation of K-RAS [9]. This gene encodes a small GTPase that regulates cellular signaling downstream of growth factor receptors. The most common K-RAS mutations are the GAT (aspartic acid; G12D), GTT (valine; G12V), and TGT (cysteine; G12C) mutations at codon 12 [10]. Instead, among the suppressor genes that are commonly mutated in PDAC there are P16/CDKN2A, TP53, and SMAD4/DPC4 [11]. 

However, subsequent mutational changes are necessary and take place gradually over time, exacerbating the genomic instability. 

Besides genetic background, risk factors for PDAC are strongly related to age [12], obesity, diabetes [13], chronic pancreatitis [14], and lifestyle, especially cigarette smoking [15]. 

Macroscopically, PDAC appears as a solid and firm white yellowish poorly defined mass [5]. It is rarely diagnosed at early stages and presents at diagnosis as an advanced lesion (pT2 or higher), with infiltration of surrounding structures (peripancreatic adipose tissue, duodenum, stomach, portal vein). The reasons for this include the frequent lack of symptoms and the proximity of major blood vessels, which increases chances of early invasion by the tumor [16].

Several PDAC variants can be recognized based on distinctive histological characteristics, associated with specific genetic signatures (Appendix A). Distinct non-invasive precursor lesions of PDAC have also been identified, providing further insight into PDAC carcinogenesis (Appendix A). Key components of PDAC are invasive and mucin-producing epithelial lesions that microscopically appear structured as irregular tubular glands, most frequently when PDAC is in the proximal pancreas. 

Improved diagnosis of PDAC precursor lesions would allow timely intervention before progression to invasive PDAC.

## 3. The Vital Crosstalk between PDAC and Tumor Microenvironment

Cancer is described as a hetero-cellular system containing both neoplastic and stromal cells that progressively form a surrounding protective microenvironment (tumor microenvironment, TME) where tumor development is facilitated.

A characteristic feature of PDAC is the so-called “desmoplastic reaction”, an abundant fibrotic response whereby atypical tumor glands are embedded within a prominent desmoplastic stroma. This fibrotic reaction creates a barrier that can prevent the penetration of chemotherapeutic agents and promote cancer growth. 

Different pancreatic stromal cells, such as pancreatic stellate cells (PSCs), cancer associated-fibroblasts (CAFs), immune cells such as tumor-associated macrophages (TAMs), regulatory T cells (Tregs), and myeloid-derived suppressor cells (MDSCs) cooperate to create a typical highly immunosuppressive stroma [17]. Consequently, the survival of cancer becomes strongly related to the intense crosstalk with the TME, which is now established as a dynamic entity (Figure 1) [18].

The TME has an active role also in metastasis. PC is often considered a metastatic disease from the time of clinical diagnosis. The most common metastatic site for PC is the liver, followed by the lungs, peritoneum, and bones [19]. Several components of the tumor microenvironment, such as PSCs and CAFs, can participate to the process of escaping of cancer cells from the primary site: they start to generate additional extracellular matrix, thereby increasing tumor pressure and diminishing vascularization [20].

Consequently, this hypoxic condition may exacerbate the malignant phenotype, suggesting that a hostile niche could favor specific cell–stroma interactions through paracrine signals and extracellular vesicle (EVs) trafficking. The latter, including exosomes, microvesicles (MVs), and apoptotic bodies, are secreted by cells in the extracellular matrix as cellular messengers to “educate” the recipient cells by delivering their specific cargo [21].

In this context, Leca et al. have shown that, under stressful conditions, CAFs within the PDAC microenvironment could manipulate cancer cell aggressive behavior through the secretion of specific ANXA6+-extracellular vesicles [22]. In turn, PDAC cells may uptake those EVs carrying the annexin A6 and so increase their spreading and metastatic potential. 

In addition, PSC-derived exosomes can promote neoplastic cell proliferation by delivering specific nuclear material. Li et al. have reported that PSC-derived exosomes deliver miR-5703 into PC cells that in turn binds and suppresses CMTM4 [23]. They also proposed that miR-5703 serum levels could be a promising diagnostic biomarker in PC.

On the other hand, the tumor itself can shape the surrounding environment to prepare the so called “soil” in secondary sites through the formation of pre-metastatic niches [24]. 

Finding new undiscovered ways of communication between stroma and cancer may provide novel therapeutic strategies for tumor tracking.

## 4. The “Dark Side” of Cell Death in PC Cells

### 4.1. Necroptosis: A Highly Specific Way of Cell Death

Cell death resistance is a recurrent feature of most types of cancer, including PDAC [25]. The different ways of cell dying can be broadly classified in two major categories: “regulated” and “accidental” cell death [26]. Regulated cell death (RCD) is an autonomous mechanism controlled by genes which can act in two different scenarios. In physiological settings (e.g., embryonic development, adult tissue homeostasis) in the absence of exogenous stimuli, RCD, also known as programmed cell death (PCD), controls the maintenance of cell populations and tissue turnover [27]. On the other hand, RCD can also be triggered by exogenous perturbations as an adaptive response, during immune reactions or when cells are damaged by diseases or harmful agents [28]. The currently known spectrum of RCD types includes apoptosis, autophagy-dependent cell death, pyroptosis, ferroptosis, necroptosis, parthanatos, entosis, NETosis, lysosome-dependent cell death (LCD), alkaliptosis, and oxeiptosis. Among them, apoptosis is the most common and well-studied process for its key role in regulating cell fate [29]. Furthermore, resistance to apoptosis induction results in enhanced cell viability and insensitivity to chemo- and radiation therapies [30]. 

Unlike RCD, accidental cell death (ACD) immediately occurs in extreme microenvironmental conditions that cause the uncontrollable cellular demise, ending up with a characteristic inflammatory reaction that specifically differentiates it from RCD [31]. ACD manifestations resemble morphological features of necrosis, including swelling of cytoplasm and organelles, especially mitochondria, followed by the explosive release of molecules to the extracellular space, upon loss of plasma membrane integrity. For this reason, it is commonly assumed that necrosis always occurs in accidental settings [32]. For example, cancer cells can respond to stresses, such as hypoxia and nutrient deficiency, by undergoing necrosis, thus considered as a form of “reparative cell death” [33]. 

Despite the fact that necrosis is usually defined as an accidental and uncontrollable event, segments of this response were recently recognized to be regulated by defined molecular pathways, leading to coin the new term “necroptosis”.

Necroptosis has been described as a novel form of RCD, a sort of “chimera” born from apoptosis and necrosis [34] and characterized by a necrotic cell death morphology in response to activation of a defined and programmable mechanism. Cells generally undergo swelling and membrane rupture, with a subsequent release of intracellular content. 

Although the necroptotic process can be triggered by extracellular or intracellular activators, the TNFα-induced necroptosis is the best characterized necroptotic pathway. TNFα is a pro-inflammatory cytokine that generally induces an inflammatory response though the downstream activation of NF-κB signaling. After the binding of TNFα to the tumor necrosis factor receptor type I (TNFR1), a high molecular weight complex, named complex I, initially arises after the recruitment of the following proteins: TNFR1-associated death domain protein (TRADD), receptor interacting serine/threonine kinase 1 (RIPK1), TNF-receptor-associated factor 2 (TRAF2), cellular inhibitors of apoptosis (cIAP1 or cIAP2), and linear ubiquitin chain assembly complex (LUBAC) [35]. The activation of the NF-κB signaling is then allowed by the upstream ubiquitination of RIPK1, a component of the complex I.

Interestingly, depending on the microenvironmental stimuli, a destabilization of complex I is responsible for the initiation of both apoptosis and necroptosis. Complex I can undergo conversion into complex IIa where the activated caspase-8, together with FADD and TRADD, executes apoptosis. Instead, the inhibition of RIPKI ubiquitination causes translocation of complex I into complex IIb, composed of RIPK1, FADD, and caspase-8. In this condition, complex IIb induces apoptosis through activated caspase-8. However, inhibition of caspase-8 can lead to the shift of the RIPK1-mediated death process from apoptosis to necroptosis, via the assembling of RIPK1, RIPK3, and mixed lineage kinase like (MLKL) into a new complex known as necrosome [36].

The formation of necrosome, a complex consisting of RIPK1, RIPK3, and mixed lineage kinase like (MLKL), is necessary for the execution of necroptosis. The activation of RIPK3 leads to the phosphorylation of MLKL. Phosphorylated MLKL shifts from its monomeric state to an oligomeric active state and then translocates from the cytosol to the inner leaflet of the plasma membrane. Here, the oligomerized and activated MLKL has been proposed to directly disrupt membrane bilayer integrity through the binding to phosphatidylinositol lipids and cardiolipin [37]. This interaction, allowed by a patch of positively charged residues in the 4HB domain of MLKL, seems to promote membrane rupture directly by forming channels and pores or indirectly by interacting with ion channels flux into cells [38,39]. However, the theory that Ca^2+^ influx is required for necroptosis downstream of MLKL was recently denied by the study of Ros et al., who also showed that necroptotic plasma membrane rupture is mediated by osmotic forces [39], indicating that contribution of Ca^2+^ influx to necroptosis is context-dependent. Following membrane disruption, a rapid inflammatory response occurs through the release of cytokines and damage-associated molecular patterns (DAMPs). 

### 4.2. Necroptosis Is a “Double-Edged Sword” in Cancer

Necrotic foci at tumor sites are a frequent occurrence resulting from inadequate vascularization and subsequent metabolic stresses, such as glucose deprivation and hypoxia. 

On the other hand, necroptosis is not a simple passive consequence of triggering events. It is a finely regulated way of dying that neoplastic cells can choose to activate or switch off, according to induction pathways. 

The role of necroptosis in oncology remains controversial, due to heterogeneous data derived from studies conducted across several distinct human cancers. An analysis of the results to date highlights context-dependent opposite functions, i.e., tumor suppression and tumor promotion, for this cell death program.

It has been reported that the susceptibility to necroptosis by several human cancers could be explained by the tendency to genetically disrupt Caspase-8 activation [36]. Within most cancers, genes coding for pro-apoptotic caspases, such as CASP8, are often under genetic deletion pressures, leading to inactivation of apoptotic machinery. Interestingly, it has been found that necrosome-related genes (such as RIPK1, RIPK3, or MLKL) experience unspecific genetic deletions at the same rate as background genetic aberration frequency, seen in cancers [36]. However, the hypothesis that caspase-8 deletion could be associated with compensatory deletion of the necroptotic pathway remains to be demonstrated. 

Several cases of decreased expression of many key modulators of necroptosis have been found in different tumors, suggesting that cancer cells may evade necroptosis as it happens within apoptosis. Nugues et al. have identified a significant reduction in RIPK3 expression in acute myeloid leukemia samples [40], while in colon cancer tissues both RIPK1 and RIPK3 appeared to be downregulated versus adjacent normal tissue [41]. RIPK1 and RIPK3 switching off was proven to be associated with worse prognosis and disease progression both in patients with breast cancer [42] and in those with head and neck squamous cell carcinoma [43].

Some studies have also suggested the possibility that pro-necroptotic genes could be under epigenetic control such as methylation-driven DNA [44].

The key to this analysis is the inflammatory background to which necroptosis actively participates by secreting several pro-inflammatory cytokines and chemokines. During this process, necroptotic cells also liberate a plethora of damage-associated molecular patterns (DAMPs), promoting a final immunogenic outcome that in turn facilitates tumor-suppression. Yang et al. reported that TNF- and chemotherapy-driven necroptosis in cancer cells can release specific DAMPs such as ATP and HMGB1, thereby promoting the activation of tumor-regressive anticancer immunity [45]. The release of DAMPs from dead cells was found not sufficient by itself to trigger robust cross-priming of cytotoxic CD8^+^ T cells, which requires RIPK1 signaling and NF-κB-induced transcription in the dying cells [46], consistent with a model of anticancer action associated with pro-necroptotic processes.

However, the downregulation of necroptotic factors may not occur in all cancers: high expression of RIPK1 or MLKL has been reported to predict worse prognosis for some cancer patients, both in melanoma [47] and breast tumors [48]. Consistent with these observations, RIPK1 is commonly found overexpressed in glioblastoma, correlating with a poorer outcome [49]. Moreover, Ando et al. recently reported that RIPK3 and MLKL are highly expressed in human pancreatic tumors compared to normal pancreas [50]. These findings challenged the idea of necroptosis as an only fail-safe mechanism to prevent cancer when apoptosis is inhibited, highlighting a potential alternative tumor-promoting role. It should be noted that depending on the cell type, necroptotic cells can release a variety of chemokines, such as CCL2, CXCL8/IL8, CXCL1/2, and CSF2. These molecules can facilitate the recruitment of myeloid cells and granulocytes and consequently promote a tumor-associated immunosuppression [36].

Thus, the modulation of anticancer immunity driven by necroptosis could take a specific direction according to several factors such as the tumor type, the microenvironment, and the immune cell types involved. From this perspective, immunosurveillance mediated by DAMPs may more easily occur in “hot tumors” as compared to “cold tumors”, which are characterized by the lack of T cell infiltration. 

These “two faces” of necroptosis can be observed in the same tumor. He et al. classified colon cancer patients into two necroptosis-related molecular subtypes with diverse clinical outcomes and tumor microenvironment infiltration characteristics. Based on the differentially expressed genes between the two molecular subtypes, they found a high necroptosis risk signature score (NRS-score) associated with poor prognosis and especially with an immunosuppressive microenvironment, whereas colon cancer patients with low NRS-score showed a better overall survival [51]. These data support the idea that necroptosis may also shield tumors from antitumor immune responses by fostering an immune escape mechanism [52].

Indeed, when blocking necroptosis of tumor cells through MLKL deletion, lung metastases were significantly inhibited [46]. Consistent with this, tumor cells were discovered to induce programmed necrosis of endothelial cells, with the consequence of facilitating neoplastic cell extravasation [53].

Taken together, these findings raise the issue that the outcome of necroptosis in cancer may be tissue-context dependent. In this perspective, the immunosuppressive TME of PC may turn out to be a decisive element to consider necroptosis as a tumor-promoting factor.

### 4.3. Necroptosis and PDAC: Friends or Foes?

Due to the desmoplastic reaction, PDAC belongs to the category of “cold tumors” which are characterized by a low infiltration of immunogenic cells, especially effector T cells, in contrast to the extensive presence of immune-suppressive cells, such as CD4^+^, CD25^+^ Tregs, MDSCs, and TAMs. This immuno-evasive background can have a relevant impact on the behavior of pancreatic cancer cells: they may jealously preserve the immunosuppressive nature of the milieu by activating many pathways and, in this perspective, it cannot be excluded that they also undergo necroptosis. All these events would occur in a significant inflammatory scenario that is characteristic of PC. Here, necroptotic cells could give their contribution by producing additional inflammation-inducing factors, helping to establish a chronic inflammatory state that consequently may stimulate tumor growth.

Interestingly, a dual function of necroptosis may reflect that attributed to inflammation: it is known that the activity of inflammatory cells combined with the type and level of inflammation-modulating factors control the balance between their pro- and antitumor effects. Thus, while in the initial stage of cancer growth, the acute phase responses may show anticancer action; in a chronic inflammatory state, the presence of inflammatory cells, especially in the TME, acts for the benefit of cancer cells, stimulating their survival and proliferation [54]. In the same way, when massive acute necroptosis happens following chemotherapy or irradiation treatment, the anti-tumor immunity is enhanced through the activation of cytotoxic CD8^+^ T cells; in contrast, a chronic necroptosis would have the opposite effect of an improved immune-suppression by producing molecules that modulate MDSC and M2-like macrophages [55]. 

Both scenarios have been observed in PC [56]. Considering the advantageous effects of chemotherapy in promoting cancer cells apoptosis, PDAC-bearing mice were initially treated with gemcitabine for a similar impact on necroptosis, thereby confirming that chemotherapy increased expression of RIPK1/3 in vivo. The second step was that of studying the effects of RIPK3 deletion both in vitro and in vivo: surprisingly, while in vitro K-RAS^G12D^-transformed PDAC cells increased their proliferative rate upon RIP3 deletion, in vivo findings were totally in contrast: p48Cre;KrasG12D;Rip3^−/−^ pancreases showed a diminished rate of acinar replacement by dysplastic ducts, and a slower PanIN progression when compared to p48^Cre^;Kras^G12D^(KC);Rip3^+/+^ pancreases, indicating that RIPK3 deletion protects against oncogenesis.

To confirm the hypothesis of tumor-promotion by necroptosis, it has been shown that RIPK3 deletion inhibits infiltration by TAMs and MDSCs. This occurred via a concomitant decrease in the expression of chemokine attractant CXCL1 and macrophage inducible Ca^2+^-dependent lectin receptor (Mincle), proposed to mediate the pro-tumorigenic immune suppression associated with RIPK3 signaling [56]. Thus, we can suppose that the opposite outcome of RIPK1/3 expression in in vivo and in vitro assays must be a function of the presence and the type of the microenvironment, well defined in PDAC. 

Taken together, these data support the idea of necroptosis as a “friend” of PDAC, as it tends to boost its growth and development by maintaining an inflammation-related immunosuppressive microenvironment (Figure 2).

However, there is a need to evaluate these findings, especially in clinical trials since studies have been conducted only in vitro and through animal models.

Questions remain also about the “timing” of this programmed process. Jiao et al. indicate that necroptosis only happens in the late stage of tumor development in breast cancer [48]. Under stressful conditions, the expression of key necroptosis mediators seems to be reprogrammed by cancer cells to restore the necroptotic machinery [52].

As changes in TME can have a critical effect in metastasis, necroptosis may have an additional role in remodeling the surrounding environment to promote neoplastic seeding. The idea that necroptosis could facilitate metastasis of PC is embraced by Ando et al. [50], after observing that conditioned media derived from necroptotic PC cells induced PC cell migration and invasion. The CXCL5 chemokine was upregulated in conditioned media, suggesting its possible implication in promoting colonization via the receptor CXCR2 [50]. 

Recently, it has been found that also necroptotic cells can produce EVs that may play roles in the necroptosis-induced immune responses [57]. This phenomenon could additionally improve the communication with the TME, inducing stromal cells to create a more favorable environment for cancer survival.

Thus, given the variety of molecular mechanisms and events supported by experimental evidence, it can be argued that necroptosis may not have a pre-defined role [36], but rather a tissue-specific one, where the stress degree as well as the amplitude and duration of inflammation may be the decisive tiebreakers of necroptosis functional fate.

## 5. Future Perspectives and Applications of Necroptosis

### 5.1. Pro-Necroptotic Markers: A Novel Focus of PDAC Research?

The lack of specific biomarkers for early detection of PDAC is a critical problem in terms of diagnosis. Although many overexpressed mRNAs or proteins in PC have been reported [58], a critical step remains—that of finding potential markers that can be easily detected in liquid biopsies with high sensitivity and specificity.

To date, the carbohydrate antigen CA19.9 represents the main serological marker associated with PC. Also known as Sialyl Lewis A, CA19.9 is a cell surface glycoprotein complex, synthesized in the normal pancreatic parenchyma and biliary tract. CA19.9 increased expression is seen not only in PC but also in stomach, colorectal, lung, thyroid, and biliary cancer [59]. 

High CA19.9 serum level is reported both in malignant and benign conditions such as pancreatitis, pancreatic cysts, diabetes mellitus, liver fibrosis, benign cholestatic diseases, and other urological, pulmonary, and gynecological diseases [60]. Thus, CA19.9 serum level cannot be used for initial diagnosis but rather for post-therapy monitoring [61].

The need of innovative design for PDAC clinical trials is growing, due to the high degree of diversity emerging from several PDAC cases, which differ not only in histopathological features but also in genetic landscape [62]. 

Aharon and colleagues discovered heterogeneity in mutation rates between early- and average-age-onset PDAC patients: SMAD4 and PIK3CA were more frequently mutated in the early-onset compared with the average-age-onset disease, suggesting the possibility to delineate a unique profile for younger patients [63].

Of particular interest, increased risk of PDAC has been associated with germline loss-of-function mutations in BRCA1 and BRCA2, as 4 to 7% of patients with PC have a family history of BRCA mutations [64]. BRCA1/2 mutations have two types of relevance. The first one is in terms of prediction and diagnosis, because their identification as predictive biomarkers allows family members to have genetic counseling [65]. The second one is linked to the field of personalized medicine, because BRCA-related tumors have been observed to have higher sensitivity to poly (ADP-ribose) polymerase (PARP) inhibitors and platinum-based chemotherapies [66].

In this perspective of implementation of new targets for PC, the idea of monitoring the expression profile of the major necroptotic biomarkers could be an additional strategy to differentiate the large number of cases.

Immunohistochemistry assays for in vivo necroptosis detection rely on the use of antibodies that recognize the main actors of the process, RIP1K, RIPK3, and MLKL in the activated form that includes phosphorylation in specific sites [67].

As regards to markers in liquid biopsies, necroptotic EVs should be involved in this analysis, with the aim to clarify their role in tumor context. 

Moreover, the development of small chemical molecules that can target necroptotic factors may represent a powerful means of exploring this pathway of cell death to find new players of the process. Then, considering the hypothesis of the pro-tumorigenic role of necroptosis, the idea of preventing the necrosome formation by using specific chemical inhibitors directed to necrosome components could also become another interesting strategy to treat cancer disease. Thus, several RIPK1 and RIPK3 are currently under clinical development for treatment of different types of cancer, while no MLKL inhibitor has been advanced into clinical trials [35]. More interestingly, HSP90 (heat shock protein 90) inhibitors, including 17AAG (ref: NCT00117988) and IPI-504 (ref: NCT00121264), have also been involved in clinical trials in patients with different cancer types, after the discovery that HSP90 may be essential for stabilizing necroptotic proteins [35,68]. 

It would be also interesting to translate findings obtained in colon cancer by He et al., as mentioned above, to PC [51]. The goal should be that of trying to classify PDAC patients into different necroptosis-related molecular subtypes according to diverse clinical outcomes and tumor microenvironment infiltration characteristics. Expression data for the main pro-necroptotic markers, RIPK1 and RIPK3, could help us to understand more about the stage of tumor, paving the way to more individualized and effective anti-cancer treatment strategies. This research is currently undergoing in our laboratories (unpublished data).

### 5.2. Immunotherapy: Is There Still a Chance?

The principal basis of cancer immunotherapy is to activate patient’s T cells to kill cancer cells, by the identification of tumor-associated antigens. Among the different strategies of immunotherapy there are the use of monoclonal antibodies, cancer vaccines, immune checkpoint inhibitors, and immune modulators. 

The last frontier of immunotherapy is currently the chimeric antigen receptor (CAR) therapy, an emerging and promising approach to redirect T cells from cancer patients into tumor-specific killer cells [69]. Once removed from the patient’s blood, T cells are engineered in vitro to express artificial receptors, called CAR, which can recognize a specific tumor-associated antigen [70].

Despite validation for hematologic malignancies [71], CAR-T therapy in solid tumors has to face some additional obstacles that involve the immunosuppressive TME, the minimal migration and persistence of CAR-T cells within the tumors, and the paucity of ideal targets [72]. Their ineffectiveness is linked in part to CAR-T cell exhaustion in TME [73], a phenomenon that happens also in pancreatic cancer, as it was studied by the group of Good et al. [74]. In the general perspective of immunotherapy, it can be expected that necroptosis of cancer cells could facilitate the activation of immune responses. However, as discussed above, the necroptotic immunobiology is contextually variable and depends on the tumor immune landscape. The desmoplastic reaction surrounding PDAC cells promotes T-cell capture, preventing them from reaching cancer cells [75]. As an effect, activated necroptosis may become an additional enemy to fight, given its potential in exacerbating the infiltration of immune-suppressive cells.

It is also important to consider the phenotype of tumor-associated macrophages (TAMs) found in pancreatic TME: while MHCIIhiTNFα+ M1-like macrophages promote anti-tumor cytotoxic T lymphocyte (CTL) activity by recruiting Th1 cells [56], CD206 + IL − 10 + M2-like macrophages enhance the expansion of Th2 cells and Tregs in PDAC [76]. 

Consistent with this, Wang et al. have recently shown that RIPK1promotes tolerogenic macrophage differentiation in the pancreatic cancer tumor microenvironment [77]. They tested the potential effects in vivo of RIPK1 inhibition in mice and in organotypic models of human PDAC, thereby finding that RIPK1 inhibitors could reprogram TAMs toward an immunogenic M1-like phenotype and induce adaptive immune protection through cytotoxic T cell activation and T helper cell differentiation. Furthermore, RIPK1 inhibition also enabled efficacy for checkpoint receptor (PD-1) and co-stimulatory ligand (ICOS)-based immunotherapies [77]. Based on this evidence, clinical trials for PDAC using RIPK1 inhibitors such as GSK3145095 (ref:NCT03681951) are currently under development. 

Thus, to validate immunotherapeutic approaches in PC, the current mission is that of exploring alternative strategies in the search for new targets in the tumor microenvironment that may contribute to disrupt this physical barrier for immune actors. 

A consistent strategy may be that of the “normalization of the stroma”: the idea is to turn abnormalities in the tumor stroma, which are involved in tumor resistance to immunotherapy, into a normal-like context, by targeting some drivers of chronic hypoxia and chronic inflammation [78]. For example, retrospective observational cohort studies found that patients with PC, who were already taking angiotensin-converting enzyme inhibitors because of preexisting cardiovascular disease, showed longer survival [79]. This group of blockers, such as losartan, acts by inhibiting the TGF pathway, which is a key activator of fibroblasts associated with the development of fibrosis in cancer [80]. 

An additional goal would be that of reprogramming activated CAFs into a dormant state [81], as in the case of Froeling et al. who used the all-trans retinoic acid to induce quiescence in fibroblasts, obtaining reduced expansion and increased apoptosis of PDAC cells in murine models [82].

Looking for several strategies to disrupt the communication between cancer cells and stroma, that is hypothesized to be mediated also by necroptotic cells released in PDAC, is still one of the main attractive challenges [83].

## 6. Conclusions

PC remains one of the most devasting neoplastic diseases with poor prognosis and limited options for successful therapy. The complex genetics, the metastatic potential and the extensive TME that grows around the tumor primary site, combined with the lack of early diagnostic markers and specific symptoms, have made this cancer an unstoppable, terrible force.

A way to better reduce the high resistance of PDAC to therapy is by trying to penetrate its desmoplastic reaction/TME. An improved understanding of the mechanisms that contribute to maintain secure the PDAC, among which necroptosis, will be fundamental to achieve this goal.

By collecting data from recent works, necroptosis is increasingly emerging as an unexpected source for the tumor to improve its interaction with the surrounding microenvironment, especially with cellular components that may help neoplastic cells to suppress the immune system. Our review supports that this revolutionary tumor-promoting function attributed to neoplastic cells should be explored in the context of pancreatic cancer, where the support given by the TME is decisive for cancer survival, development, and spreading. Recent advances in immune and targeted therapy may inspire novel strategies that could in turn increase the efficacy of precision medicine, thereby improving the outcome of different subtype-specific patients. In this perspective, a multidisciplinary approach arises as the one and only way for oncology research to translate findings into concrete and innovative clinical tools and to try to become stronger in this constant battle against PC.

## Figures and Tables

**Figure 1 ijms-24-12633-f001:**
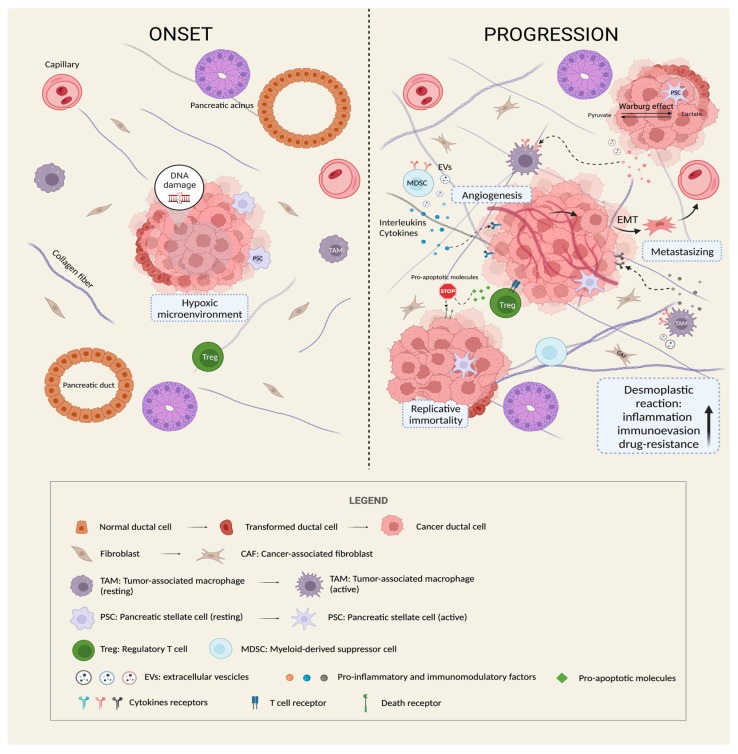
The role of TME in the progression of PDAC. As shown in the figure, once PC arises, it enhances aggressiveness and malignancy by inducing a series of modifications of the surrounding microenvironment. A reciprocal interplay is established between cancer cells and other TME cellular types (CAFs, PSCs, Tregs, TAMs, and MDSCs) through the release of extracellular vesicles (EVs) and pro-inflammatory and immunomodulatory paracrine factors. As a result, the pancreatic stroma becomes denser and more compact (desmoplastic reaction), the degree of inflammation intensifies, and the normal immune response of the body is weakened, thus creating a “protective barrier” that promotes tumor immune evasion and drug resistance, and provides a perfect niche for growth, vascularization, and metastatic spreading. Furthermore, it is worth noting that cancer cells undergo metabolic reprogramming (Warburg effect) and acquire the ability to replicate indefinitely and to resist to pro-apoptotic stimuli thanks to the progressive accumulation of DNA mutations (genomic instability), especially of oncogenes and tumor suppressor genes. Created with BioRender.com.

**Figure 2 ijms-24-12633-f002:**
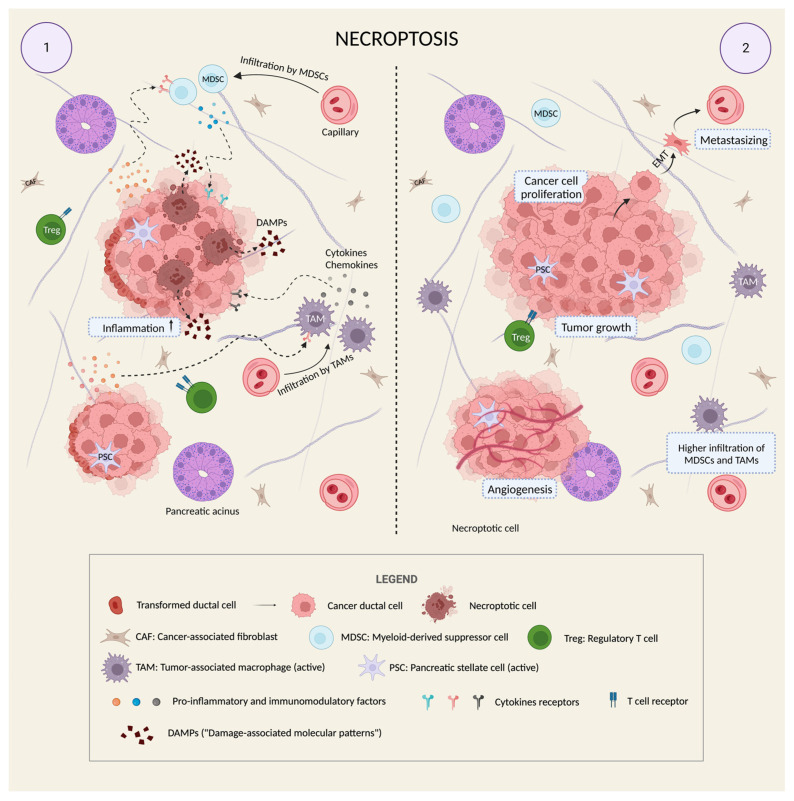
Necroptosis as an additional means of tumor progression in PDAC. PC may take advantage of a particular cell death mechanism that is able to activate the tumor microenvironment to enhance tumor malignancy. Panel 1: Within an already present desmoplastic reaction that is evident from the fibrous and dense stroma and from the activation of the typical cellular types of the tumor microenvironment (PSCs, CAFs, MDSCs, and TAMs), some cancer cells differ for a characteristic morphological change: these are necroptotic cells (brown cells in panel 1). They have a completely altered internal architecture, but their distinctive feature is the disruption of the plasma membrane which occurs in the form of pores in the membrane itself. Through these pores, the cytoplasm is violently released in the extra-cellular space. Among the wide variety of released material, the DAMPs (“damage-associated molecular patterns”), which include ATP, heat-shock proteins, and other molecules, can increase the degree of inflammation in the microenvironment and make this inflammation chronic. Cytokines and chemokines released by cancer cells are particularly important due to their ability to target some of the cellular types of the microenvironment (especially MDSCs and TAMs), thereby regulating their activity from anti-tumoral to pro-tumoral. Panel 2: As a result, thanks to the combination of microenvironment activation and necroptotic death, PDAC creates the perfect conditions to grow, promote angiogenesis, metastasize, and escape the immune response of the body, also generating a self-feeding cycle which supports its inflammatory background and its viability (the increasing infiltration of MDSCs and TAMs because of necroptotic stimuli is made evident in the cartoon). Created with BioRender.com.

## Data Availability

Not applicable.

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
