# Peer review of "The Dual Role of Necroptosis in Pancreatic Ductal Adenocarcinoma"

_ijms, 2023, doi:10.3390/ijms241612633_

Round 1
Reviewer 1 Report (New Reviewer)
The manuscript “The dual role of necroptosis in pancreatic ductal adenocarcinoma” submitted by Giansante et al. mainly deals with the role of necrotic/necroptotic cell death in the course of the initiation, progression, and therapy of pancreatic ductal adenocarcinomas. This review introduces pancreatic adenocarcinomas (PDACs) and its tumor microenvironment, then shortly necroptosis as one of the regulated cell death modalities, and lastly necroptotic signaling in PDACs with its effect on tumor growth including possible future perspectives. The outcome view of necroptotic signaling in PDACs favors in contrast to apoptosis its pro-inflammatory, metastases-promoting role potentially interfering with current anti-cancer therapy of PDAC. In general, this review provides a good overview of the current knowledge of necroptotic signaling in pancreatic carcinoma. However, there are still some points that should be addressed:
1. The overview of the mechanism of initiation, progression, and execution of necroptosis as a necrotic mode of regulated cell death (RCD) is presented very superficially, and for readers not familiar with necroptotic signaling might be instrumental introducing different modes of the initiation of necroptosis including a relevant figure, regulatory nodes of necroptosis including MLKL itself and final steps in the execution of necroptosis leading to bursting of necroptotic cells caused by MLKL-induced pores in cellular membranes and disturbance of cellular homeostasis.
2. Do not freely mix necrosis and necroptosis – necroptosis is RCD (not programmed cell death – this term is predominantly used for apoptotic cell death during embryogenesis) with necrotic outcome (similarly as other necrotic modes of RCD such as pyroptosis or ferroptosis).
3. It would be also helpful to introduce and discuss possible pharmaceutic interventions for blocking necroptosis in PDAC.
Author Response
Please see the attachment below.

Reviewer 2 Report (New Reviewer)
This review by Giansantle et al. concisely summarizes the current knowledge about the relationship between pancreatic ductal carcinoma and necroptosis, a recently identified regulated cell death. The authors show the positive and negative roles of necroptosis in the development of PDCA depending on the stages of tumors and the extent of necroptosis. This manuscript will be suitable for publication in Int J Mol Sci. However, the followings are specific comments to further improve the manuscript.
1. Lines 218-219: The authors mentioned “High expression of RIPK1 or MLKL has been reported to be worse prognosis some cancer patients, both in melanoma and breast tumors.” However, the authors did not mention about the cohort studies of PDCA to compare the disease-free survival based on the expression levels of necroptosis-specific genes, RIPK3 and MLKL. If such studies have been already published, these information would be very informative.
2. Lines 185-186: The authors did not correctly explain the mechanisms how MLKL induces necroptosis. Please change the manuscript according to the following suggestions.
1) The authors need to mention that phosphorylated MLKL undergoes oligomerization, and then translates to the plasma membrane.
2) It is currently believed that MLKL does not form a rigid pore structure like GSDMD, but induces membrane rupture.
3) Moreover, contribution of Ca2+ influx to membrane rupture may be context-dependent, since other study did not support their results (Cell Rep. 19, 175-187). Therefore, contribution of Ca2+ influx to necroptosis appears to be context-dependent.
Author Response
Please see the attachment below.

Round 2
Reviewer 1 Report (New Reviewer)
The authors addressed all issues raised in the former version of the manuscript and thus I do recommend accepting it for publication in the IJMS.
This manuscript is a resubmission of an earlier submission. The following is a list of the peer review reports and author responses from that submission.
Round 1
Reviewer 1 Report
Short Summary
Manuscript titled: “The double face of Necroptosis in Pancreatic Ductal Adenocarcinoma (PDAC) “ offers a concise survey of main types of PDAC, based on pathohistological characteristics, with some references on genetic and protein alterations. Further on, authors discuss the importance of tumor microenvironment and alterations in cell types with the emphasis on cross-communication, involving extracellular vesicles (EV), which enables cancer progression. The process of necroptosis is discussed in detail. Finally, authors consider perspectives for necroptosis research in terms of the discovery of novel biomarkers for diagnosis/prognosis, novel drug targets and possibilities for immunotherapy of PDAC.
Manuscript is scientifically sound, with clear structure and relevant, up-to-date literature. Authors should rephrase conclusion in order to emphasize and summarize scientific issues in a concise manner. English language editing required.
Minor issues
Line 276 - “Furthermore, in comparison with PDAC, the prognosis is even worse [46]. “ (adenosquamous carcinoma ?).
Please clarify, since you stated in previous sentence that adenosquamous carcinoma is a variant of PDAC.
Suggestion: add ‘classical PDAC’
Line 327 - “… ectopic pancreatic liver tissue in the pancreas. They are extremely uncommon but associated with a very poor prognosis...“
Term pancreatic should be omitted, to exclude ambiguity
Line 330 - Table 1, entry No. 3 “UCOGC” in the ‘PDAC variants’ column are not explained in the text, nor in the legend. It is not clear which variant is described; instead of CC – full name Colloid carcinoma should be given, with abbreviation in parenthesis Also, Table 1 legend should include all abbreviations
Line 403 - “the capability to survive of neoplastic cells”
should be changed to: the capability of neoplastic cells to survive.
Line 461 - “…PDA cells can increase the release of alanine …”
PDA cells – abbreviation meaning? PDAC cells?
Line 516 - “choice to activate”
Should be changed to: choose to activate
Line 559 - necroptosis risk score signature (NRS-score)
Should be: Necroptosis Risk Signature score
Lines 603-604 - “Both situations have been observed in pancreatic cancer by the central study of Seifert et al. “
Add Rererence No.
Lines 682-701- Figure 2 panel 2 should be mentioned/ commented in the legend
Line 840 - “as a main actor of immune response dynamics”
Should be changed to: as a main FACTOR.
Author Response
Please see the attachment below.

Reviewer 2 Report
The manuscript has many grammatical errors and uses in my opinion at times language that is not appropriate for scientific writing.
Some examples below:
Line 27: “and a lot of questions remain” I believe that this is not appropriate language for a Review abstract. Please rewrite the sentence.
Line 49-50: “interested by tumor: approximately the 70% of pancreatic” please delete “the”
Line 53: “Instead, patients with body and tail pancreatic cancers has more nonspecific” have - not has
Line 68-71: According to this, probably, the fact that this tumor is significantly aggressive must be assigned to specific unexpected strategies developed by neoplastic cells, that we think to be enhanced by the same pancreatic tissue context, particularly hostile but at the same time necessary to cause a more powerful response of the tumor itself.
The authors should be more specific, this is a vague statement. Explain in depth: significantly aggressive (what does this mean?), to specific unexpected strategies developed by neoplastic cells (what are these strategies?), that we think to be enhanced (who thinks that?)
Line 73: “In this perspective, what is supposed by many research groups”
Please rewrite to appropriate scientific writing style
Line 77: “analyzing the real contribute of cell death” contribution not contribute
many more errors following this throughout the manuscript.
I am happy to fully review this manuscript once the authors have corrected all grammatical errors and the use of appropriate scientific writing style.
Author Response
Please see the attachment below.

Round 2
Reviewer 1 Report
The Reviewer would like to thank the authors for accepting suggestions and making changes.
I have no further comments.
Author Response
Please see the attachment below.
